# Systemic DNA Damage and Repair Activity Vary by Race in Breast Cancer Survivors

**DOI:** 10.3390/cancers16101807

**Published:** 2024-05-09

**Authors:** Shraddha Divekar, Ryan Kritzer, Haokai Shu, Keval Thakkar, Jennifer Hicks, Mary G. Mills, Kepher Makambi, Chiranjeev Dash, Rabindra Roy

**Affiliations:** Georgetown University’s Lombardi Comprehensive Cancer Center, Georgetown University Medical Center, Washington, DC 20057, USA; sd1292@georgetown.edu (S.D.); rykritzer@gmail.com (R.K.); hs1052@georgetown.edu (H.S.); kevalvthakkar@gmail.com (K.T.); js936@georgetown.edu (J.H.); mg266@georgetown.edu (M.G.M.); khm33@georgetown.edu (K.M.)

**Keywords:** racial disparity, CometChip assay, double-strand break repair, single-strand break repair

## Abstract

**Simple Summary:**

Non-Hispanic Black breast cancer survivors have poorer outcomes than White survivors, but the biological mechanisms underlying these disparities are unclear. We discovered novel race-based differences in systemic DNA damage and repair activity among breast cancer survivors. This finding suggests DNA damage and repair are important basic science mechanisms in cancer disparities.

**Abstract:**

Non-Hispanic Black breast cancer survivors have poorer outcomes and higher mortality rates than White survivors, but systemic biological mechanisms underlying these disparities are unclear. We used circulating leukocytes as a surrogate for measuring systemic mechanisms, which might be different from processes in the target tissue (e.g., breast). We investigated race-based differences in DNA damage and repair, using a novel CometChip assay, in circulating leukocytes from breast cancer survivors who had completed primary cancer therapy and were cancer free. We observed novel race-based differences in systemic DNA damage and repair activity in cancer survivors, but not in cells from healthy volunteers. Basal DNA damage in leukocytes was higher in White survivors, but Black survivors showed a much higher induction after bleomycin treatment. Double-strand break repair activity was also significantly different between the races, with cells from White survivors showing more sustained repair activity compared to Black leukocytes. These results suggest that cancer and cancer therapy might have long-lasting effects on systemic DNA damage and repair mechanisms that differ in White survivors and Black survivors. Findings from our preliminary study in non-cancer cells (circulating leukocytes) suggest systemic effects beyond the target site, with implications for accelerated aging-related cancer survivorship disparities.

## 1. Introduction

Genome stability, DNA damage, and DNA repair are associated with cellular aging and are major hallmarks of breast cancer and other cancers [1,2,3,4]. Accelerated cellular aging results from adverse social and metabolic risk factors over the life course. It is recognized as a central mechanism leading to cancer, and adverse outcomes in cancer survivorship [5]. Standard cancer treatments, particularly radiation therapy and chemotherapy, often involve agents or strategies that damage the DNA in non-cancerous tissues, resulting in systemic effects associated with accelerated aging processes [5,6]. Previous studies on cells obtained from breast cancer patients have shown a higher oxidative stress level, susceptibility to DNA damage, and a decreased DNA repair capacity, than cells from healthy controls [3,4,7,8]. Several specific repair mechanisms repair single-strand and double-strand breaks in DNA [9]. However, it is unclear whether these changes are also seen systemically, e.g., in circulating leukocytes and other non-cancer cells, as the characterization of these repair mechanisms at the systemic level during the cancer survivorship period is still not well defined.

The incidence of breast cancer is slightly higher in non-Hispanic White (NHW) women compared to non-Hispanic Black (NHB) women, but mortality from breast cancer is 27% higher in NHB than NHW women [10]. There are currently over 4 million breast cancer survivors in the U.S. who, despite a generally favorable prognosis, face lifelong risks of clinically important symptoms related to poor quality of life and adverse health effects, such as obesity, metabolic syndrome, and diabetes [10]. There are well-documented disparities in the prevalence of these symptoms and metabolic comorbidities, with NHB breast cancer survivors bearing a higher burden than NHW women [10]. Systemic mechanistic pathways related to accelerated cellular aging possibly underlie persistent race/ethnic differences in cancer mortality, due to disparities in these clinical symptoms and adverse metabolic health effects, but have been understudied. Recent studies have shown that like NHW women, NHB women carrying germline pathogenic mutations in DNA damage response (*ATM*, *CHEK2*) and repair genes (*BRCA1*, *BRCA2*, *PALB2*, *RAD51D*, *XPB*, *FANCC* and *RECQL*) are at moderate to high risk for breast cancer [11]. However, the expression of DNA repair genes, primarily in single strand break/base excision repair (SSBR/BER) and double strand break repair (DSBR), differed in breast tumor tissue by race [12]. Given the integral role of DNA damage and repair in accelerated aging, [13] and recent findings of race differences in tumor tissue [11,12], systemic differences in these mechanisms should be investigated in NHB and NHW breast cancer survivors.

Alkaline single cell gel electrophoresis (SCGE), also known as Comet Assay, is routinely used to measure DNA damage and repair activity due to its high sensitivity and simplicity [14,15,16]. However, an issue with the standard comet assay is its reproducibility [17]. CometChip Assay employs the same fundamental concepts as SCGE; but utilizes a microwell system that traps ~300, non-overlapping, single cells within each well of a 96-well plate [17,18]. Thereby, this innovative CometChip assay provides a high-content, highly sensitive, quantitative, and high-throughput DNA damage assay platform, with high reproducibility. The use of the CometChip assay to measure DNA damage has potential in breast cancer translational research [19,20]. The versatile nature of the CometChip assay allows the detection of specific DNA damage types by modifying the alkaline CometChip assay, to detect and quantify diverse damage types, such as single-strand breaks, alkali–labile abasic sites, and double-strand breaks, whereas the neutral CometChip assay predominantly detects double-strand breaks [21,22,23,24]. Double-strand breaks are repaired by non-homologous end joining/homologous recombination (NHEJ/HR) pathway, whereas the oxidized bases are repaired by BER pathway, and the bulky adducts, induced by UV-light and polyaromatic hydrocarbons, are repaired by nucleotide excision repair (NER) pathway [22,25,26,27]. Therefore, CometChip assays can be used to detect a variety of damage types, as well as to identify specific DNA repair mechanisms and pathways in clinical samples.

We used the CometChip assay to measure global double-strand damage and repair capacity in breast cancer survivors, and participants without a history of cancer (non-cancer participants). We also compared DNA damage and repair by race among breast cancer survivors to investigate potential mechanistic pathways for cancer mortality disparities between NHB and NHW women. 

## 2. Materials and Methods

### 2.1. Overview

Figure 1 describes the study workflow schematic. CometChip assays were performed on buffy coat cells isolated from peripheral blood from non-Hispanic Black (NHB) and non-Hispanic White (NHW) cancer-free participants and breast cancer survivors, to measure DNA damage at a basal level, after bleomycin (BLM) treatment and in the recovery/repair period.

### 2.2. Patient Recruitment and Sample Collection

Adult non-Hispanic Black (NHB, *n* = 13) and White (NHW, *n* = 12) invasive breast cancer survivors, with no history of breast cancer recurrence or known breast cancer-related germline mutations, who had completed primary treatment (surgery, chemotherapy, radiation therapy) at least six months, and at most three years, before study entry (women currently on hormone therapy were eligible) were recruited through community-based approaches, and identification and screening using the Survey, Recruitment, and Biospecimen Collection Shared Resource (SRBSR) at Georgetown University’s Lombardi Comprehensive Cancer Center. After providing written informed consent, all participants completed a demographic and medical history questionnaire, and anthropometrics measurements to determine weight and height. Notably, the majority of breast cancer patients undergo testing for germline mutations, particularly focusing on BRCA mutations. We identified non-BRCA patients from the electronic medical record (EMR) database and included them in this study. It is also worth noting that patients who reported non-BRCA mutations may not have undergone formal genetic testing. Venous blood (20 mL) was collected into sodium heparin tubes. All data collection activities were conducted at the Lombardi Cancer Center’s Office of Minority Health and Health Disparities Research. The Georgetown MedStar Institutional Review Board approved the study protocol (#STUDY00003904). 

All participant blood specimens were transported to Lombardi Cancer Center’s Tissue Culture and Bio-banking Shared Resource (TCBSR) and processed within two hours of collection. Processing was performed under the National Cancer Institute’s (NCI) Best Practices for Biospecimen Resources guidelines. Aliquots were stored, and viable leukocytes were isolated in the form of buffy coat by centrifugation and cryopreserved in aliquots at −80 °C freezers with emergency power backup. We also procured buffy coat samples of 6 NHB and 6 NHW cancer-free participants, matched on age-intervals from SRBSR at Georgetown University’s Lombardi Comprehensive Cancer Center. Previous studies have demonstrated the reliability and validity of Comet assays on stored samples [28,29].

### 2.3. CometChip Assay for DNA Damage Detection and Repair Kinetics Evaluation

Alkaline and Neutral CometChip assays were conducted following previously published procedure [17] and using a polydimethylsiloxane (PDMS) stamp, with an array of micro pegs (a kind gift from Dr. Engelward’s lab at MIT, Cambridge, MA, USA), to form an array of about 300 microwells in each of 96 macro wells on the agarose chip. CometAssay Alkaline (biotechne, catalog no. 4256-010-CC, Minneapolis, MN, USA) and Neutral (biotechne, catalog no. 4257-010-NC) reference cells served as baseline controls representing DNA damage at various levels in alkaline and neutral CometChip assays. These standard control cells are a reference for monitoring the variation in day-to-day CometChip assay procedures for evaluating patient samples. This was based on recommendations from consensus statements published in 2020, and updated in 2023 [30,31], by an international group of Comet assay users. We added these reference undamaged and damaged cells as internal controls for every alkaline and neutral assay run and for every human sample we analyzed. This was to make sure that the observed DNA damage and repair activity variation is actually inherent to the sample, and not due to assay variability. The reproducibility of the experimental protocol across seven runs for alkaline assays and four runs for neutral assays is presented in Appendix A. The mean moments of four internal controls from the experimental runs were within ±3 SD of the mean moment of all individual experiments for CC0 (0.48 ± 0.31), CC2 (1.23 ± 1.26), NC0 (0.97 ± 0.26), and NC2 (9.53 ± 5.18), suggesting high reproducibility. Alkaline and Neutral CometChip assays were conducted with few modifications from previously published protocols, and are described below in the appropriate sections.

#### 2.3.1. Cell Recovery and Preparation for CometChip Assays

On the day of the experiment, the aliquoted Buffy coat cells and control cells were thawed in a 37 °C water bath and kept on ice. The Buffy coat cells and the control cells were then resuspended in 2.5 mL and 0.5 mL cold Phosphate Buffered Saline (PBS), respectively. HCT 116 cells were freshly harvested from cell culture plates on the day of CometChip assay, washed with cold PBS, resuspended in 2.5 mL cold PBS, and stored on ice until loaded into the CometChip.

#### 2.3.2. CometChip Preparation

For the Comet Chip assay, 96-well agarose gels were prepared by using molten 1% normal melting point (NMP) agarose (ThermoFisher, Waltham, MA, USA) in PBS. The molten agarose was poured onto the hydrophilic side of a Gel Bond^®^ film (Lonza, Walkersville, MD, USA), which was placed up in a rectangular petri dish lid and evenly spread across the lid. The polydimethylsiloxane (PDMS) stamp, with an array of micropegs, was gently placed on top of the gel. The gel was allowed to solidify for 15 min. The PDMS was removed, and 10 mL PBS was added to the dish. The micropore formation was confirmed by observing the gel on an inverted microscope with 10× magnification. The excess gel from the lid was removed, and the gel attached to the Bond^®^ film was submerged in PBS and stored before use at 4 °C for no longer than a week. 

#### 2.3.3. CometChip Cell Loading

The whole CometChip experiment was carried out in yellow light to minimize spurious DNA damage to the cells. The agarose gel attached to the Bond^®^ film was placed on a glass plate, and a bottomless 96-well plate (VWR) was pressed on the agarose chip to form 96 macrowells. The whole sandwich was then secured with the binder clips (staples) on four sides. Each macrowell contained an array of about 300 microwells. Each microwell (30 µm) receives mostly a single cell. Rarely, more than one cell is deposited in a microwell. Two or more cells, if deposited, are discarded manually during Comet analysis by Comet Assay Software (comet analysis is described below in Section 2.3.6). One hundred microliters of single cell suspension (~2000 or more cells) were added into each macrowell, and the sandwiches with the chip were incubated in a 37 °C cell culture incubator in the presence of 5% CO_2_ for 30 min. After incubation, the media from the wells was aspirated and the agarose chips were washed with 10 mL PBS to remove excess cells. This process results in generation of array of single cells in microwells (Figure 2A). Molten 1% low melting point (LMP) agarose (ThermoFisher; kept at 46 °C until use) was poured on the chips. To ensure complete solidification, the gels were kept at room temperature for 3 min and then at 4 °C for 5 min.

#### 2.3.4. Bleomycin and Methyl Methanesulfonate Treatment for Repair Assay

Bleomycin (BLM; Cayman Chemical Company, catalog no. 13877, Ann Arbor, MI, USA) is a radiomimetic agent that induces double-strand breaks along with single-strand breaks containing 3′-phosphoglycolate/5′-phosphate ends or 4′-oxidized abasic sites in DNA [32,33]. Doses of BLM (2.5–10 µg/mL for alkaline and 1.25–5 µg/mL for neutral assays) were prepared in RPMI 1640 media, and the patient cells embedded in CometChips were submerged in BLM-containing media and incubated for 15 min in a 37 °C incubator to induce DNA damage. Optimization of BLM dose and repair time for alkaline and neutral assays are shown in Appendix A. Based on the normal repair response of the cells where the damage level reached the basal or near basal level, we selected 2.5 µg/mL BLM for damage induction and a maximum of 60 min for repair kinetics under alkaline conditions to analyze all survivors’ samples. Similarly, we chose 5 µg/mL BLM for damage induction and a full 120 min for repair kinetics for neutral conditions.

The CometChips were cut into pieces for the convenience of their use for multiple BLM doses and different repair time points. Post-BLM treatment, the chips were washed briefly three times by submerging them in PBS. After optimization of the doses and the repair time points, patient samples were analyzed routinely for damage induction and repair kinetics by incubating them with BLM concentrations of 2.5 μg/mL and 5.0 μg/mL for alkaline and neutral assays, respectively. For repair kinetics, 0–60 min and 0–120 min were chosen for alkaline and neutral assays, respectively. Basal DNA damage in patient cells and damage in reference Trevigen control cells were analyzed immediately after loading cells and overlaid with LMP agarose in a cold lysis solution (10 mM Tris-HCl, 2.5 M NaCl, 100 mM Na_2_EDTA, 1% *v*/*v* Triton X-100 at pH 10) for 18 h at 4 °C for alkaline CometChip assay, whereas the chips were submerged in 43 °C pre-warmed lysis solution (10 mM Tris-HCl, 2.5 M NaCl, 100 mM Na_2_EDTA, 1% N-Lauroylsarcosine, 10% DMSO, 0.5% *v*/*v* Triton X-100 at pH 9.5) and incubated for 18 h at 43 °C for the neutral CometChip assay. To analyze induced DNA damage, the post-BLM treated CometChips were washed with PBS and placed immediately in either a cold lysis buffer or 43 °C pre-warmed lysis solution, and processed following the remaining steps on alkaline or neutral CometChip assays. To evaluate repair kinetics, the post-BLM treated CometChips were washed with PBS, placed immediately in a growth medium containing RPMI 1640 and 10% FBS, and incubated for aforementioned repair time points for alkaline and neutral assays, respectively, in a 37 °C cell culture incubator in the presence of 5% CO_2_. At the completion of each repair time point, the growth media was aspirated and the CometChip or its fragments were placed in CometChip assay-specific lysis buffer and processed as described above.

To assess the damage type, SSB, or DSB detection by alkaline and neutral CometChip assays under our assay conditions, we used an alkylating agent, methyl methanesulfonate (MMS; ThermoFisher, catalog no. H55120.06), as it does not form DSBs, but forms SSBs indirectly. It induces methylated bases such as 3-methyl adenine and 7-methyl guanine in DNA. SSBs are formed upon spontaneous hydrolysis of 7-methylguanine and cleavage of the resulting alkali labile abasic sites in alkaline CometChip assay. We used HCT116 (ATCC catalog no. CCL-247, RRID: CVCL_0291) cells for these optimization experiments, because they are a well-established model for studying DNA damage and repair processes, and they have been widely used in previous studies, including ours [34,35,36,37]. The HCT116 cell line was obtained directly from Lombardi Cancer Center’s Tissue Culture and Biobanking Shared Resource (TCBSR) and fingerprinted for identity confirmation. The cell line was passaged for 3 months after receipt for use in the described experiments. The cell line was routinely tested for the presence of mycoplasma, with the latest test on 18 January 2024, using the MycoFluor Mycoplasma Detection Kit according to manufacturer’s instructions (Molecular Probes, catalog no. M-7006). The HCT116 cells embedded in CometChips were submerged in DMEM media containing 1 mM MMS and incubated for 1 h in a 37 °C incubator to induce DNA damage. Post-MMS treated CometChips were washed with PBS, placed immediately in either cold lysis buffer or 43 °C pre-warmed lysis solution, and processed following the remaining steps on alkaline or neutral CometChip assays.

#### 2.3.5. Alkaline and Neutral Electrophoresis

For the alkaline assay, the gel was washed with cold distilled water and denatured to unwind nuclei in an alkaline unwinding buffer (0.3 M NaOH and 1 mM Na_2_EDTA at pH 14) for 40 min at 4 °C, followed by electrophoresis in the same alkaline unwinding buffer at 4 °C for 30 min at a constant 21 V and ~300 mA, using the Trevigen Comet Assay^®^ Electrophoresis System II (Biotechne, catalog no. 4250-050-ES). After electrophoresis, the gel was washed briefly by submerging it in the neutralization buffer (0.4 M Tris-HCl at pH 7.5) for 5 min at room temperature, it was then kept in the same buffer for 15 min at 4 °C, and finally stored in the same buffer at 4 °C for up to 2 days. For the neutral assay, the gel was washed and incubated in cold neutral electrophoresis TBE buffer (90 mM Tris-HCl, 90 mM Boric acid, 2 mM Na_2_EDTA at pH 8.5) for 60 min at 4 °C, followed by electrophoresis in the same TBE buffer at 4 °C for 15–16 min at a constant 21 V and 8–12 mA, using the same Trevigen Comet Assay^®^ Electrophoresis System II.

#### 2.3.6. Gel Staining, Imaging and Comet Analysis

The gels were stained for 20 min on a shaker with 1× SYBR Gold (ThermoFisher, catalog # S11494) in TE buffer, and, followed by brief washing, imaged using an automated fluorescent microscope (Keyence BZ-X710 series in one) at 4× magnification in the GFP channel (488 nm). The images of the comets were captured by automatic scanning, compressed, and stitched for analysis and necessary editing by the Comet Assay Software (Trevigen, catalog no. 4260-000-CS), and a .csv file was generated for graphical and statistical analysis. Figure 2B shows an example of the damaged and undamaged cells during imaging and analysis.

### 2.4. Data Analysis, Statistics, and Reproducibility

The graphs were generated using GraphPad Prism v10. Differences in DNA damage (Mean ± SD) between untreated and MMS/BLM treatment (Figure 2C), and between NHB and NHW (Figures 4A,B and 5A,B), were assessed using unpaired two-tailed *t*-test with Welch’s correction and unpaired, two-tailed Mann–Whitney test, respectively (significance level *p* < 0.05). A two-way repeated measures ANOVA (significance level *p* < 0.05) was used to assess the impact of disease (Figure 3A,B) and race (Figures 6A–D and 7A–D) across different repair time points. Further analyses, to investigate the role of covariates on DNA damage and repair activity, were conducted using generalized linear models, adjusted for BMI, age, cancer stage, and treatment. All statistical analyses were performed using GraphPad Prism v10 and R v4.2. No statistical methods were used to pre-determine sample sizes for this pilot study, and the normality assumption was not formally tested. Data collection and analysis were performed blinded to the experimental conditions. Pre-determined criteria guided participants’ eligibility for this study. We did not exclude any data points from our analyses. All analyses are from five replicates of each measurement for each participant’s sample.

## 3. Results

### 3.1. CometChip Assay Optimization for Global DNA Damage and Double-Strand Break, Bleomycin Dose, and Repair Kinetics

This pilot study investigated race-based differences in systemic basal DNA damage, damage induction, and repair after bleomycin (BLM) treatment using a high-sensitivity, high-throughput novel CometChip assay in circulating leukocytes from cancer-free participants and breast cancer survivors [17] (Figure 1). All cancer survivors had completed primary treatment at least six months before sample collection and were cancer-free at enrollment. Buffy coat cells isolated from venous blood were collected from 13 NHB and 12 NHW breast cancer survivors. Participants reported a mean age of 58 years at enrollment and a mean BMI of 30.6 kg/m^2^. Distributions of breast cancer stage, histology, smoking status, prior history of other cancers, and treatment were well-matched between the two races, with 67% of NHW and 62% of NHB women having an early-stage diagnosis. All participants were non-smokers and had invasive cancer. None of the participants had prior history of other cancers (Appendix A).

Through optimization, we confirmed that the CometChip assay, under alkaline conditions detects, global DNA damage, including alkali-labile sites, double-strand breaks (DSB), and single-strand breaks (SSB). In contrast, under the neutral condition, only DSBs are detected (Figure 2C). Both alkaline and neutral comet assays were used to investigate cancer-free controls vs. cancer survivors and race-based differences in moment, %DNA in tail, and proportion of undamaged cells at baseline and different repair time points (0, 15, 30, 60, 120 min) following BLM treatment.

### 3.2. Repair of BLM-Induced Double-Strand Break and Global DNA Damage in Cancer-Free Women and Breast Cancer Survivors

We observed differences in the DNA repair capacity of DSBs in leukocytes between cancer-free women and breast cancer survivors (Figure 3A). Although most cellular damage from BLM has been repaired by the 15-min repair point for cancer-free women and breast cancer survivors, there are significant differences in repair kinetics and residual damage between the two groups. Compared to cancer survivors, there seemed to be increased sustained damage and subsequent repair activity 15 min post-BLM among cancer-free participants. Repair activity was different between cancer-free women and breast cancer survivors, as measured by the interaction of two groups with repair activity (mean moment; overall (*p* < 0.01), Figure 3A). Global DNA damage and repair differences between these two groups were similar to DSB. However, the differences in magnitude between cancer-free participants and cancer survivors were lower, with less likelihood of statistical significance across the four measures (Figure 3B).

### 3.3. Race Effect on Basal and Damage Susceptibility and Repair of BLM-Induced Double-Strand Break and Global DNA Damage in NHB and NHW Breast Cancer Survivors

Mean (± SD) basal DNA damage in leukocytes was higher in NHW women compared to NHB women for both DSB (3.7 ± 1.43 au for NHW vs. 1.26 ± 0.81 au for NHB, *p* < 0.01; Figure 4A) and global damage (2.24 ± 0.66 au for NHW vs. 1.06 ± 1.2 au for NHB, *p* < 0.01; Figure 5A). However, NHB-derived cells showed a much higher induction (normalized to basal levels) in response to BLM than NHW cells for both damage types (DSB: 3.34 ± 2.95 au for NHW cells vs. 8.12 ± 5.6 au for NHB cells, *p* = 0.02; Figure 4B; global damage: 0.94 ± 0.76 au for NHW vs. 7.64 ± 6.6 au for NHB, *p* = 0.01; Figure 5B). In addition, variability in basal damage and damage induction was higher for NHB than NHW individuals.

We observed differences in the DNA repair capacity of DSBs in leukocytes between NHB and NHW groups (Figure 6A–D). Although most cellular damage from BLM has been repaired by the 15-min repair point for NHW and NHB groups, there are significant differences in repair kinetics and residual damage between the two races. Among the NHB group, there did not seem to be any measurable ongoing damage or repair activity after 15 min, compared to the sustained damage and subsequent repair activity among the NHW group, even at 120 min post-BLM. Repair activity was different between the NHB and NHW groups, as measured by the interaction of race with repair activity (mean moment; overall (*p* = 0.01) and in damaged cells (*p* = 0.01), Figure 6A,C). Consequently, the proportion of undamaged cells across repair time points was higher in NHB cells than in NHW cells (*p* < 0.01, Figure 6D). However, at 120 min post-BLM, the proportion of undamaged cells in the NHW group was similar to basal levels. Still, among the NHB group, it did not recover to the basal levels (Figure 6D). In generalized linear models, race differences remained significant after adjusting for age and BMI. However, repair differences were attenuated after further adjustment for cancer stage and treatment, suggesting a systemic effect of cancer and cancer therapy that persists beyond the treatment period. 

Global DNA damage and repair differences between NHB and NHW survivors were similar to DSB. However, the magnitude of race differences was much lower and less likely to be statistically significant across the four measures (Figure 7A–D). We also investigated DSB damage and repair in 12 cancer-free women (6 NHW women and 6 NHB women) to determine whether the differences in cancer patients might result from race differences, irrespective of cancer status. We found no differences in DNA damage or repair across time points among NHB and NHW women without cancer (Appendix A). However, the sample size is small to draw any definitive conclusions.

## 4. Discussion

Our study using the novel CometChip assay is one of the first to show novel differences in systemic DNA damage and repair, primarily DSB repair, between NHB and NHW breast cancer survivors. Although some studies have reported race-based differences in systemic DNA damage and repair activity in healthy volunteers, none have focused on cancer survivors [38]. It is also important to note that prior molecular mechanism studies investigating racial differences were either not focused on DNA repair mechanisms, or were conducted only among cancer patients, not in cancer survivors [12,39,40,41,42].

Our results on systemic DNA damage and repair activity between NHB and NHW survivors add to recent literature that showed differences in the expression of DNA repair genes, especially those involved in DSBR, in breast tumor tissue by race [12]. Our results of comparing breast cancer survivors to cancer-free women also support findings from Scuric et al., who used DNA damage assays on circulating WBCs that suggest the persistence of systemic effects of cancer and cancer treatment, primarily radiation therapy and chemotherapy, well beyond the end of treatment [43]. Contrary to our preliminary hypothesis, NHB survivors had lower basal DNA damage and lower measured DNA damage post-BLM. They also had less sustained DSB repair activity after the BLM challenge than NHW cells. NHW cells returned to basal proportions of undamaged cells after post-BLM repair activity, but NHB cells did not (Figure 6D). The observed low repair activity could be due to mutations in BRCA genes in those survivors. However, it is important to note that we selected BRCA mutation-free survivor samples for this study. Moreover, we did not force the leukocytes in our assay to proliferate by treatment with any mitogen, such as phytohemagglutinin [44], and we speculate that NHEJ, which does not require BRCA, was possibly the predominant DSBR pathway in our BLM-treated cells.

Race-based differences in systemic DNA damage and repair activity could also be related to signaling pathways involved in accelerated aging, such as inflammation and oxidative stress, associated with cancer and cancer treatment [45]. It is noteworthy that some leukocytes damaged by chemo or radiotherapy will not necessarily survive for extended periods, but some do. In addition, the treatment-related changes in systemic mechanisms (e.g., inflammation and oxidative stress) might be long-lasting, and continuously damage healthy cells, including new leukocytes [43]. Persistent treatment-associated changes may also alter immune functions and cause post-treatment health complications in the survivors. In a mouse study, restraint-induced stress activated genes responsible for priming the T cells to either undergo apoptosis or proliferation, the major function needed for cellular immunity [46].

The strengths of our study include the quantitative measurement of different damage types and repair pathways, rather than only measuring global damage (alkaline comet assays), as is commonly reported [8,20]; innovative data analyses, such as segregating cells into undamaged (zero damage) and damaged; and the focus on systemic DNA damage and repair in the survivorship period, rather than on tumor tissue/cells. Our study is the first of its kind in systemic disparities in cancer survivorship, but is limited by a relatively small sample size, lack of breast cancer subtypes and detailed cancer treatment data, and lack of survivorship-related outcome data. Additionally, although we looked at DSB, global DNA damage, and their repair, we did not measure other DNA damage and activities, such as base damage and its repair.

## 5. Conclusions

In conclusion, we report novel differences in systemic DNA damage and repair by race in breast cancer survivors. Future confirmatory studies in diverse cancer survivor populations are needed to validate our findings and to investigate the association of underlying inflammation and aging-related pathways using gene expression and mutation data, socio-environmental factors, and survivorship-related outcomes with the race-based differences observed in our study.

## Figures and Tables

**Figure 1 cancers-16-01807-f001:**
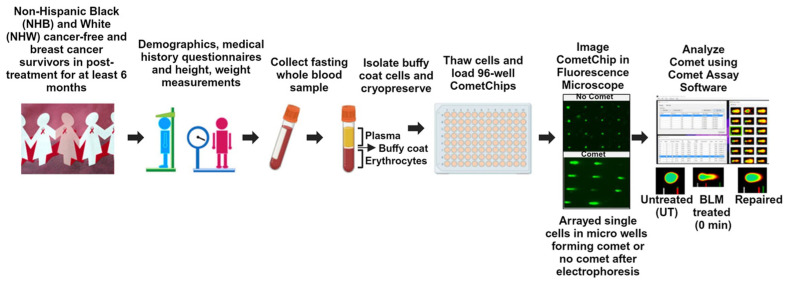
Study Schematic. For details, see Materials and Methods. Created with Biorender.com (accessed on 2 May 2024).

**Figure 2 cancers-16-01807-f002:**
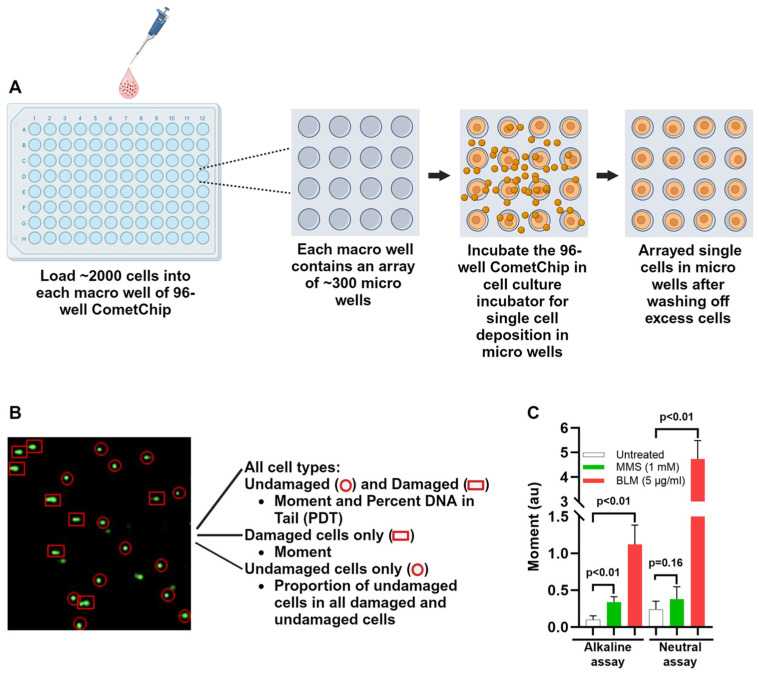
(**A**) Cell loading and single cell array generation. (**B**) A representative CometChip image shows undamaged and damaged nucleoids and the mode of analysis used in this study. (**C**) Alkaline (global DNA damage) and neutral (double-strand break) CometChip assay optimization for detection of double-strand breaks (DSB), and single-strand breaks (SSB). HCT116 cells were treated with methylmethane sulfonate (MMS) and bleomycin (BLM) and analyzed in alkaline and neutral CometChip assays. Tail Moment is used as a DNA damage parameter. *p* < 0.05 was considered significant. Created with Biorender.com (accessed on 2 May 2024).

**Figure 3 cancers-16-01807-f003:**
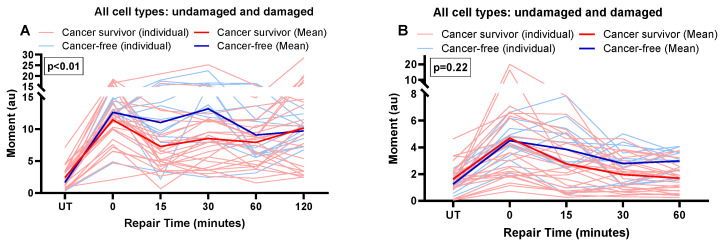
Repair of BLM-induced double-strand Break and global DNA damage in cancer-free women and breast cancer survivors. The repair kinetics of BLM-induced DNA double-strand breaks was measured by neutral (**A**) and alkaline (**B**) assays. DNA double-strand breaks (**A**) and global damage (**B**) were measured in cells after BLM treatment during their recovery period at different time points. A repeated measures ANOVA was conducted to assess the impact of disease (cancer) on repair kinetics. *p*-values represent the statistical significance of the interaction of disease with repair activity at different time points. *p* < 0.05 is considered significant.

**Figure 4 cancers-16-01807-f004:**
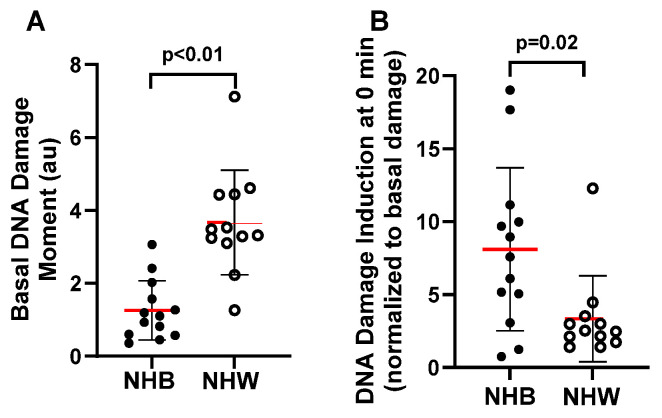
Effect of race on basal and damage susceptibility of BLM-induced double-strand break in NHB and NHW breast cancer survivors. Basal (**A**) and induction/susceptibility (**B**) of BLM-induced double-strand breaks were measured by neutral assay. Differences in DNA damage (Mean ± SD) between NHB and NHW (**A**,**B**) were assessed using unpaired, two-tail Mann-Whitney test. The red lines denote the mean value. DNA damage induction in (panel **B**) was calculated using the formula: [DNA damage (moment) after BLM treatment − basal DNA damage (Moment) before BLM treatment]/basal DNA damage (moment) before BLM treatment.

**Figure 5 cancers-16-01807-f005:**
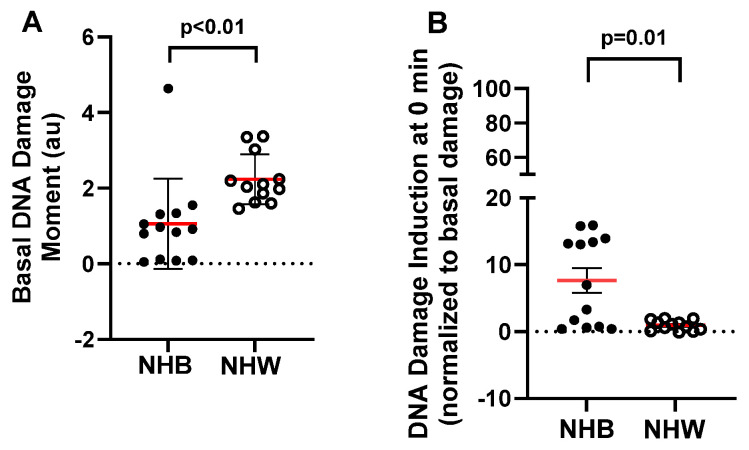
Effect of race on basal and damage susceptibility of BLM-induced global DNA damage in NHB and NHW breast cancer survivors. Basal (**A**) and induction/susceptibility (**B**) of BLM-induced global damage were measured by alkaline assay. Differences in DNA damage (Mean± SD) between NHB and NHW survivors (**A**,**B**) were assessed using an unpaired, two-tail Mann–Whitney test. The red lines denote the mean value. DNA damage induction in panel B was calculated using the formula described in Figure 4 legends.

**Figure 6 cancers-16-01807-f006:**
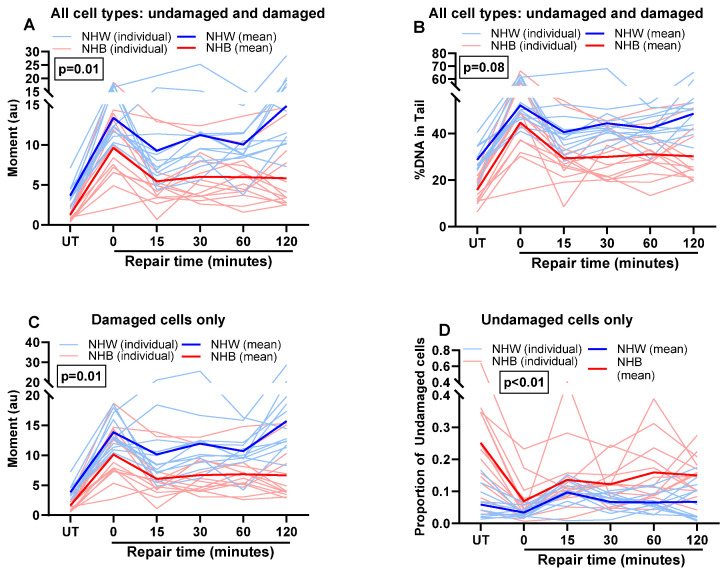
Effect of race on repair of BLM-induced double-strand break in NHB and NHW breast cancer survivors. The repair kinetics of the BLM-induced DNA double-strand breaks were measured by neutral (**A**–**D**) assay. DNA double-strand breaks were measured in cells after BLM treatment during their recovery period at different time points. A repeated measures ANOVA was conducted to assess the impact of race on repair kinetics. *p*-values represent the statistical significance of the interaction of race with repair activity at different time points. Upon segregation of all cells, damaged and undamaged cell populations were assessed by excluding cells with zero values and including cells only with zero values, respectively. *p* < 0.05 was considered significant.

**Figure 7 cancers-16-01807-f007:**
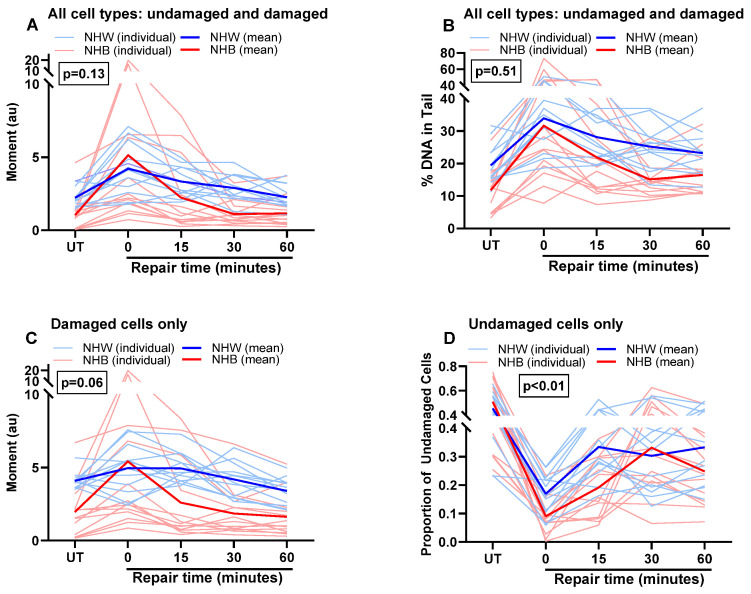
Effect of race on repair of BLM-induced global DNA damage in NHB and NHW breast cancer survivors. The repair kinetics of BLM-induced global DNA damage was measured by alkaline (**A**–**D**) assay. DNA damage was measured in cells after BLM treatment during their recovery period at different time points. A repeated measures ANOVA was conducted to assess the impact of race on repair kinetics. *p*-values represent the statistical significance of the interaction of race with repair activity at different time points. Upon segregation of all cells, damaged and undamaged cell populations were assessed by excluding cells with zero values and including cells only with zero values, respectively. *p* < 0.05 is considered significant.

## Data Availability

The data presented in this study are available upon request from the contact corresponding author. We used no accession codes, unique identifiers, or web links for publicly available datasets used in this study.

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
