# Peer review of "Systemic DNA Damage and Repair Activity Vary by Race in Breast Cancer Survivors"

_cancers, 2024, doi:10.3390/cancers16101807_

Round 1

Reviewer 1 Report (Previous Reviewer 1)

Comments and Suggestions for Authors

This manuscript has been improved and my previous concerns have been resolved. I don't have more comments about this revised version.

Author Response

Reviewer 1

Comment 1: This manuscript has been improved and my previous concerns have been resolved. I don't have more comments about this revised version.

Response: We thank the reviewer for helping us to improve our manuscript.

Reviewer 2 Report (Previous Reviewer 2)

Comments and Suggestions for Authors

The authors have addressed most or all of my original comments in this resubmission, and it has significantly improved the manuscript. The revisions are quite voluminous and detailed, indicating the authors have taken the  reviewer comments seriously and have tried to handle everything hoestly and thoroughly. There are still a few critical issues that may become more obvious now in this remission, however. 

The authors have also added material to demonstrate the principle of their assay, how the metho works in practice, and also how its analyzed. That has all been descibed in a publication from 2010 (not by the same authors) but I believe its important to describe materials and methods used  in a paper and not just refer to 14-year old manuscripts published elsewhere. Ghe authors now also describe this method in more detail, how single cells are prepared and transferred into the assay platform, and added a new figure that shows how everything works. This is a small but necessary addition. The figures 2, 3 and especially, 4, are really small and it should be considered to merge them into a single figure that has all of these elements combined. 

Maybe the most relevant addition the authors added was a control group that is analyzed in comparison to the 2 patient groups; namely, persons that never had any cancer therapy and are therefore "naive" for the questions adressed in the manuscript. This has been made into a new Figure 5. The differences between cancer survivors and controls are a bit marginal and not significant, at least in one form of the assay (alkaline lysis) and thats probably due to the high level of variaton between the patients/samples - which is a general problem with this assay. The situation is better for acidic lysis and there is also less variation. However, the following figures show that all other experiments are using the alkaline assay format - the one where there was no significant difference between untreated and formerly treated patients. So its a bit unfortunate and problematic. Adding fig. 5 may be necessary, but it still doesnt really help very much to demonstrate that we are looking into biologically relevant, robust, and reproducible differences. Reproducibility is the key issue. 

Nevertheless, Fig. 5 shows how important it is to include control groups. This should have probably also been done for the analysis shown in Fig. 7 and 9 ; but it has not been added probably because they represent  "historic" experiments that cannot be repeated.

Most importantly, however: Can you repeat the assay shown in Fig. 6 and 8 and validate your own results with a 2nd, independent set of patient samples of different race ?  The authors should at least comment on this in detail, as the robustness of the assay is still not overly convincing to me. Do you have access to a 2nd set of patients/samples for independent validation? This is the single most important and critical restriction I still have before this manuscript can be published. In most journals, this would also be the most valid parameter that needs to be demonstrated - reproducibility in independent patient sets. The total number of patients in both groups compared is still small. 

This is even more critical as the authors state themselves that the study is limited by "a relatively small sample size, lack of breast cancer subtypes and detailed cancer treatment data, and lack of survivorship-related outcome data". 

Yes, this is very much true, but therefore, the issue of reproducibility becomes even more relevant, and 12 vs 13 patients may simply not be enough to prove the point your are trying to make. 

The authors also try to explain the phenomenon they observe but - can you really explain ? Genetic differences between "racial backgrounds" are notoriously difficult to justify or validate. Tnis may need some more thorough (literature) research... 

Comments on the Quality of English Language

there are no majpr issues (as indicted above) and there are also no serious typos etc that cannot be fixed prior to acceptance

Author Response

Reviewer 2

Comment 1: The authors have addressed most or all of my original comments in this resubmission, and it has significantly improved the manuscript. The revisions are quite voluminous and detailed, indicating the authors have taken the reviewer comments seriously and have tried to handle everything honestly and thoroughly.

Response: We are happy to learn that we have addressed to all original comments of the reviewer in the resubmission, and thank the him/her for helping us to improve our manuscript.

Comment 2: There are still a few critical issues that may become more obvious now in this remission, however. 

The authors have also added material to demonstrate the principle of their assay, how the method works in practice, and also how its analyzed. That has all been described in a publication from 2010 (not by the same authors) but I believe it’s important to describe materials and methods used in a paper and not just refer to 14-year old manuscripts published elsewhere. The authors now also describe this method in more detail, how single cells are prepared and transferred into the assay platform, and added a new figure that shows how everything works. This is a small but necessary addition. The figures 2, 3 and especially, 4, are really small and it should be considered to merge them into a single figure that has all of these elements combined. 

Response: We agree with the reviewer. We have merged Figures 2, 3 and 4 in Figure 2 in the new resubmission.

Comment 3: Maybe the most relevant addition the authors added was a control group that is analyzed in comparison to the 2 patient groups; namely, persons that never had any cancer therapy and are therefore "naive" for the questions addressed in the manuscript. This has been made into a new Figure 5. The differences between cancer survivors and controls are a bit marginal and not significant, at least in one form of the assay (alkaline lysis) and that’s probably due to the high level of variation between the patients/samples - which is a general problem with this assay. The situation is better for acidic lysis and there is also less variation. However, the following figures show that all other experiments are using the alkaline assay format - the one where there was no significant difference between untreated and formerly treated patients. So, it’s a bit unfortunate and problematic. Adding fig. 5 may be necessary, but it still doesn’t really help very much to demonstrate that we are looking into biologically relevant, robust, and reproducible differences. Reproducibility is the key issue. Nevertheless, Fig. 5 shows how important it is to include control groups. This should have probably also been done for the analysis shown in Fig. 7 and 9; but it has not been added probably because they represent "historic" experiments that cannot be repeated.

Response: There is actually a significant difference (p<0.01; actual p-value = 0.001) between all Cancer-free women and Cancer survivors under neutral condition (however, it’s not acidic lysis condition). It is important to note that comet assay under neutral condition detects only DNA double-strand breaks, whereas under alkaline condition it detects a number of different DNA lesions including alkali-labile sites such as AP-sites, single-strand breaks (SSB) and also double-strand breaks (DSB). That’s why we described the alkaline comet assay for “global DNA damage” detection. We described these definitions and categorizations in our previous submissions and also in the new resubmission (lines 308-311). We, in fact, confirmed these facts in the old Figure 4 (new Figure 2C). Moreover, before we started analyzing patient samples we rigorously tested both alkaline and neutral CometChip assays for day-to-day variations and the results showed that both assays are highly reproducible (see Figure S1A). DNA damage, drug dose and repair kinetics were also optimized using control buffy coat cells for this study (see Figure S1B). Altogether, we confirmed that these assays optimized for this study are highly reproducible and were appropriate for analyzing the patient samples. The apparent variability of DNA damage response and repair among individual cancer-free and cancer survivor women are expected. That reflects their individual DNA repair biology and that has nothing to do with the assay type. Furthermore, mixture of multiple DNA damage type (not the assay method) detected under alkaline condition may have reduced the statistical significance for the difference between cancer-free vs cancer survivors (old Figure 5; new Figure 3) and Non-Hispanic White vs Non-Hispanic Black (old Figure 9; new Figure 7), albeit we do see difference in these groups. In fact, observation of these differences inspired us to discuss in the Discussion section (both in original submission and new resubmission) that in future study we need to measure other DNA damage and activities such as base damage and its repair (line 444-446). We did not test the effect of race on DNA damage and repair in cancer-free vs cancer survivors as the number of cancer-free samples for each race was low.

Comment 4: Most importantly, however: Can you repeat the assay shown in Fig. 6 and 8 and validate your own results with a 2nd, independent set of patient samples of different race?  The authors should at least comment on this in detail, as the robustness of the assay is still not overly convincing to me. Do you have access to a 2nd set of patients/samples for independent validation? This is the single most important and critical restriction I still have before this manuscript can be published. In most journals, this would also be the most valid parameter that needs to be demonstrated - reproducibility in independent patient sets. The total number of patients in both groups compared is still small. 

This is even more critical as the authors state themselves that the study is limited by "a relatively small sample size, lack of breast cancer subtypes and detailed cancer treatment data, and lack of survivorship-related outcome data". 

Yes, this is very much true, but therefore, the issue of reproducibility becomes even more relevant, and 12 vs 13 patients may simply not be enough to prove the point you’re are trying to make. 

Response: We agree with the reviewer that the “robustness and reproducibility” of the assays are foremost important. As described above in response to Comment 3, we confirmed (validated) both alkaline and neutral assays as highly reproducible for analyzing the patient samples on day-to-day basis, which is a primary requirement for clinical sample analysis (see Figure S1A). This was based on recommendations from consensus statements published in 2020 and updated in 2023 (see references 30, 31 in the manuscript) by an international group of Comet assay users.  We had a brief discussion on this issue in the original submission but have expanded on these in the Materials and Methods section (lines 134-141) in this resubmission. Moreover, the assays are optimized for drug dose and repair times for this study (see Figure S1B), and most importantly we added control undamaged and damaged cells as internal controls or reference cells for every alkaline and neutral assay run and for every human sample we analyzed. This was to make sure that the observed DNA damage and repair activity variation is actually inherent to the sample and not due to assay variability.

Our findings of cancer vs cancer-free and race-based differences are novel and hypothesis generating. We agree that these findings should be confirmed/externally validated in future large studies with larger number of samples. Unfortunately, we currently do not have the funding or access to samples to conduct these large-scale assays.

Comment 5: The authors also try to explain the phenomenon they observe but - can you really explain? Genetic differences between "racial backgrounds" are notoriously difficult to justify or validate. This may need some more thorough (literature) research... 

Response:  We do not claim or hypothesize that the observed DNA damage and repair differences in Black and White breast cancer survivors are “genetic” or based on “ancestral germline differences”. Actually, we hypothesize as presented in the introduction (lines 55-58, 63-66) and discussion (lines 426-436) that this is primarily related to cancer treatment, environmental factors, and coexistent metabolic comorbidities, like obesity; and gene-environment interactions. However, the specific reasons for the DNA damage and repair differences need to be studied in larger studies. Our study is not designed or powered to answer these questions.

Reviewer 3 Report (Previous Reviewer 3)

Comments and Suggestions for Authors

The manuscript by Devikar et al entitled “Systemic DNA Damage and Repair Activity Vary by Race in Breast Cancer Survivors” suggests that cancer and cancer therapy might have long-lasting effects on systemic DNA damage and repair mechanisms that differ in White survivors and Black survivors.  The authors demonstrated the DNA damage by Commet assay, however, the confirmation of the DNA damage by immune florescence staining using H2Ax antibody may increase the quality of the manuscript. Otherwise, the manuscript is well written, the results are well presented.

Many thanks

Author Response

Reviewer 3

Comment: The manuscript by Divekar et al entitled “Systemic DNA Damage and Repair Activity Vary by Race in Breast Cancer Survivors” suggests that cancer and cancer therapy might have long-lasting effects on systemic DNA damage and repair mechanisms that differ in White survivors and Black survivors.  The authors demonstrated the DNA damage by Commet assay, however, the confirmation of the DNA damage by immune florescence staining using H2Ax antibody may increase the quality of the manuscript. Otherwise, the manuscript is well written, the results are well presented.

Response: For this study we used previously collected cancer survivor samples with limited number of aliquots available. We used up most of the samples for our elaborate DNA repair kinetic studies with CometChip assays under multiple conditions.  Results from these studies are described in the current manuscript. Moreover, we used frozen buffy coat cells in our experiments. Fresh or frozen buffy coat cells or even whole blood cells are sufficiently good for Comet assays (see references 28, 29 in the manuscript). However, reliable cell staining with antibodies including γH2AX antibody and quantitation require PBMC (purified form of WBCs), which were not available for the patients included in this study. We propose to assay γH2AX antibody in future studies with prospective patient enrollment and prior planning of cell purification from patient blood, which is essential for reliable antibody staining. 

Round 2

Reviewer 2 Report (Previous Reviewer 2)

Comments and Suggestions for Authors

the authors have again modified the manuscript and its now better to read, the merged figures are nice,, added references are a plus... no problem. 

The main issue (for me) is the small cohort size. 

Now the authors answer (in a somewhat lengthy response) about this and I think the main point is this one:

"We agree that these findings should be confirmed/externally validated in future large studies with larger number of samples. Unfortunately, we currently do not have the funding or access to samples to conduct these large-scale assays"

Thats understandable and acceptable, and I havent expected anything else, in fact. Now the decision has to be made... accept this fact, or keep insisting in larger cohorts. 

I think its fine if the authors are stating this very clearly: they are "hypothesis-generating" and this study is NOT ultimate proof that their observation may be confirmed by others. But thats sometimes how thigs go in science. It this were another journal, rejection would be certain at this level but this is also a matter of fairness in publications, I think. The authors should get a chance to publish this but the caveat, the restrictions and limitations have to be made very clear. Then the study is acceptable for publication. 

Comments on the Quality of English Language

not an issue here

Author Response

Reviewer 2

Comment: Now the authors answer (in a somewhat lengthy response) about this and I think the main point is this one:

"We agree that these findings should be confirmed/externally validated in future large studies with larger number of samples. Unfortunately, we currently do not have the funding or access to samples to conduct these large-scale assays"

That’s understandable and acceptable, and I haven’t expected anything else, in fact. Now the decision has to be made... accept this fact, or keep insisting in larger cohorts. 

I think its fine if the authors are stating this very clearly: they are "hypothesis-generating" and this study is NOT ultimate proof that their observation may be confirmed by others. But that’s sometimes how thigs go in science. It this were another journal, rejection would be certain at this level but this is also a matter of fairness in publications, I think. The authors should get a chance to publish this but the caveat, the restrictions and limitations have to be made very clear. Then the study is acceptable for publication. 

Response: We have changed the lines 449-450 in the conclusions section of the revised manuscript to reflect the reviewer’s suggestion.

This manuscript is a resubmission of an earlier submission. The following is a list of the peer review reports and author responses from that submission.

Round 1

Reviewer 1 Report

Comments and Suggestions for Authors

In this manuscript, the authors used cell samples from cured breast cancer survivors to explore the difference in DNA damage and repair activity between the NHB race and the NHW race. Interestingly, they found the NHB has lower basal DNA damage but a much higher inducible DNA damage, this gives an important reminder that avoiding engagement of inducer during or after cancer therapy may benefit NHB individuals. Unfortunately, as the authors mentioned, conclusions here came from a relatively small sample size, and only a single element, general DNA damage/repair was considered to clarify the difference in outcomes and mortality.

As a general concern, a more detailed introduction to the CometChip assay is needed in the manuscript, rather than simple citations of literature, because most of your data came from this assay.   

Author Response

We thank you as a reviewer of our manuscript for a thoughtful critique and suggestions.  In this resubmission, we have responded point by point and made changes according to your comments (please see below). We have made the necessary revisions to address your concerns and improve the quality of the manuscript. The revisions in the manuscript text are highlighted in “gray” color.

Comment 1: In this manuscript, the authors used cell samples from cured breast cancer survivors to explore the difference in DNA damage and repair activity between the NHB race and the NHW race. Interestingly, they found the NHB has lower basal DNA damage but a much higher inducible DNA damage, this gives an important reminder that avoiding engagement of inducer during or after cancer therapy may benefit NHB individuals.

Unfortunately, as the authors mentioned, conclusions here came from a relatively small sample size, and only a single element, general DNA damage/repair was considered to clarify the difference in outcomes and mortality.

Response:  As mentioned in our discussion section we agree that sample size is relatively small but this is an exploratory study with an important finding that needs to be replicated in a future larger study.

Comment 2: As a general concern, a more detailed introduction to the CometChip assay is needed in the manuscript, rather than simple citations of literature, because most of your data came from this assay. 

Response: We agree with the reviewer. We have included additional information on CometChip assay in the Introduction (lines 67-90), and methodological information in the Materials and Methods section (lines 92-102, 147-268).

Reviewer 2 Report

Comments and Suggestions for Authors

The authors have used a "CometChip" for their analysis of DNA damage in breast cancer survivors of different racial background. The description of the use of this chip is quite detiled, including their optimization based on HCT116 colon cancer cells. The technology isnt described in much detail but refers to a paper from 2010. I would still appreciate a little figure actually showing this chip in more sdetails, especially the single-cell wells and how they are prepared; the paper isnt that voluminous and lengthy. This description, which would be helpful for the readers, could also include a brief explanation of how the imaging is done. The refered paper [16] is simply quite dated and I would assume that there have been changes to the basic protocol described in [16] since 2010. For example, I wonder how it is achieved that only single cells are loaded into these microwells. 

Then, of course, the number of patients is relatively low (13 versus 12). The statistics is certainly limited based on the small sample number and additional controls might give more confidence in the findings. For example, I dont understand why no normal control samples (= non-cancer patients) have been included, that should have been rather straightforward, to sample healthy people that never had any chemotherapy nor breast cancer. If he hypothesis of the authors is right, there should be quite obvious and strong differences in bleomycin sensitivity and the formation of DNA breaks in such control cells, and the cancer survivor cells? I also wondered if this may be discussed in the final part of the manuscript, but it wasnt. Are there plenty of papers that have shown this already (? = strong differences between cancer patients and healthy controls)  so the authors didnt even bother to include this? ANd even if there were, it would give the entire assay a different dimension, and relevance, showing that there are strong chemotherapy-induced and long-lasting effects that can be observed months or years after the therapy. 

So, in total, I do like the method, but it needs more detailed explanation, and there is room in this short manuscript. Second, I am missing the scope of the problem, and the authors should demonstrate that there are significant differences between the cancer patients (regardless of which race, probably) and the never-had-cancer controls. 

Author Response

We thank you as a reviewer of our manuscript for a thoughtful critique and suggestions.  In this resubmission, we have responded point by point and made changes according to your comments (please see below). We have made the necessary revisions to address your concerns and improve the quality of the manuscript. The revisions in the manuscript text are highlighted in “gray” color.

Comment 1: The authors have used a "CometChip" for their analysis of DNA damage in breast cancer survivors of different racial background. The description of the use of this chip is quite detailed, including their optimization based on HCT116 colon cancer cells. The technology isn’t described in much detail but refers to a paper from 2010. I would still appreciate a little figure actually showing this chip in more details, especially the single-cell wells and how they are prepared; the paper isn’t that voluminous and lengthy. This description, which would be helpful for the readers, could also include a brief explanation of how the imaging is done. The referred paper [16] is simply quite dated and I would assume that there have been changes to the basic protocol described in [16] since 2010. For example, I wonder how it is achieved that only single cells are loaded into these microwells. 

Response: The basic protocol used in this paper is unchanged from the original one published in 2010 by the MIT group. The MIT group have been publishing manuscripts using the same protocol as detailed in our revised manuscript [1-3]. In the revised manuscript we have elaborated extensively all the essential steps of the CometChip assay including microwell (single-cell wells) preparation (lines 155-185) and imaging technique (lines 258-266). We have also added a new Figure (Figure 2) showing how arrays of single cells were generated in the microwells.

 Comment 2:  Then, of course, the number of patients is relatively low (13 versus 12). The statistics is certainly limited based on the small sample number and additional controls might give more confidence in the findings. For example, I don’t understand why no normal control samples (= non-cancer patients) have been included, that should have been rather straightforward, to sample healthy people that never had any chemotherapy nor breast cancer. If he hypothesis of the authors is right, there should be quite obvious and strong differences in bleomycin sensitivity and the formation of DNA breaks in such control cells, and the cancer survivor cells? I also wondered if this may be discussed in the final part of the manuscript, but it wasn’t. Are there plenty of papers that have shown this already (? = strong differences between cancer patients and healthy controls) so the authors didn’t even bother to include this? And even if there were, it would give the entire assay a different dimension, and relevance, showing that there are strong chemotherapy-induced and long-lasting effects that can be observed months or years after the therapy. 

So, in total, I do like the method, but it needs more detailed explanation, and there is room in this short manuscript.

Second, I am missing the scope of the problem, and the authors should demonstrate that there are significant differences between the cancer patients (regardless of which race, probably) and the never-had-cancer controls. 

Response: We agree with the reviewer and have included detailed explanation and methodology of CometChip assay (see response to Comment 1 above).  We also agree with the reviewer about the importance of comparing cancer survivors with cancer-free controls. We have added this comparison in the results section (lines 315-334) with a new Figure (Figure 5).

References:

  1. Ngo, L.P.; Owiti, N.A.; Swartz, C.; Winters, J.; Su, Y.; Ge, J.; Xiong, A.; Han, J.; Recio, L.; Samson, L.D.; et al. Sensitive CometChip assay for screening potentially carcinogenic DNA adducts by trapping DNA repair intermediates. Nucleic Acids Res 2020, 48, e13, doi:10.1093/nar/gkz1077.
  2. Ngo, L.P.; Kaushal, S.; Chaim, I.A.; Mazzucato, P.; Ricciardi, C.; Samson, L.D.; Nagel, Z.D.; Engelward, B.P. CometChip analysis of human primary lymphocytes enables quantification of inter-individual differences in the kinetics of repair of oxidative DNA damage. Free Radic Biol Med 2021, 174, 89-99, doi:10.1016/j.freeradbiomed.2021.07.033.
  3. Tay, I.J.; Park, J.J.H.; Price, A.L.; Engelward, B.P.; Floyd, S.R. HTS-Compatible CometChip Enables Genetic Screening for Modulators of Apoptosis and DNA Double-Strand Break Repair. SLAS Discov 2020, 25, 906-922, doi:10.1177/2472555220918367.

Reviewer 3 Report

Comments and Suggestions for Authors

I reviewed the manuscript by Divekar et al., entitled “ Systemic DNA Damage and Repair Activity Vary by Race in Breast Cancer Survivors“

The authors present data demonstrating that the DNA damage  and DNA repair mechanisms are race dependent mechanisms. Thereby, can be used as an indicator for cancer disparities. The manuscript is well written, and the data are interesting in best form. However, the confirmation of DNA damages/DSBs by immune fluorescence staining using H2Ax is required .

Many thanks

Comments on the Quality of English Language

NA

Author Response

We thank you as a reviewer of our manuscript for a thoughtful critique and suggestions.  In this resubmission, we have responded point by point and made changes according to your comments (please see below). We have made the necessary revisions to address your concerns and improve the quality of the manuscript. The revisions in the manuscript text are highlighted in “gray” color.

Comment: I reviewed the manuscript by Divekar et al., entitled “Systemic DNA Damage and Repair Activity Vary by Race in Breast Cancer Survivors “. The authors present data demonstrating that the DNA damage and DNA repair mechanisms are race dependent mechanisms. Thereby, can be used as an indicator for cancer disparities. The manuscript is well written, and the data are interesting in best form. However, the confirmation of DNA damages/DSBs by immune fluorescence staining using H2Ax is required.

Response: γH2AX is often used as a marker for DSB detection. Histone H2A.X is a variant histone that becomes phosphorylated in response to DSBs. Current evidence suggests that γH2AX does not always indicate the presence of DSB [1], even in response to ionizing radiation [2]. The Neutral Comet assay is an authentic method to detect DSBs in genomic DNA. Thus, we used this method for detection of DSBs in clinical samples with or without challenging cells to DNA damaging agents. However, as suggested by the reviewer we would use γH2AX as an alternative marker for DSB in a larger study.

References:

  1. Soutoglou, E.; Misteli, T. Activation of the cellular DNA damage response in the absence of DNA lesions. Science 2008, 320, 1507-1510, doi:10.1126/science.1159051.
  2. Tu, W.Z.; Li, B.; Huang, B.; Wang, Y.; Liu, X.D.; Guan, H.; Zhang, S.M.; Tang, Y.; Rang, W.Q.; Zhou, P.K. gammaH2AX foci formation in the absence of DNA damage: mitotic H2AX phosphorylation is mediated by the DNA-PKcs/CHK2 pathway. FEBS Lett 2013, 587, 3437-3443, doi:10.1016/j.febslet.2013.08.028.